# Recent Development of the Molecular and Cellular Mechanisms of Hydrogen Sulfide Gasotransmitter

**DOI:** 10.3390/antiox11091788

**Published:** 2022-09-10

**Authors:** Jianyun Liu, Fikir M. Mesfin, Chelsea E. Hunter, Kenneth R. Olson, W. Christopher Shelley, John P. Brokaw, Krishna Manohar, Troy A. Markel

**Affiliations:** 1Department of Surgery, Section of Pediatric Surgery, Indiana University School of Medicine, Riley Hospital for Children at Indiana University Health, Indianapolis, IN 46202, USA; 2Department of Physiology, Indiana University School of Medicine—South Bend, South Bend, IN 46617, USA

**Keywords:** gasotransmitter, persulfidation, reactive sulfur species, reactive species interactome

## Abstract

Hydrogen sulfide has been recently identified as the third biological gasotransmitter, along with the more well studied nitric oxide (NO) and carbon monoxide (CO). Intensive studies on its potential as a therapeutic agent for cardiovascular, inflammatory, infectious and neuropathological diseases have been undertaken. Here we review the possible direct targets of H_2_S in mammals. H_2_S directly interacts with reactive oxygen/nitrogen species and is involved in redox signaling. H_2_S also reacts with hemeproteins and modulates metal-containing complexes. Once being oxidized, H_2_S can persulfidate proteins by adding -SSH to the amino acid cysteine. These direct modifications by H_2_S have significant impact on cell structure and many cellular functions, such as tight junctions, autophagy, apoptosis, vesicle trafficking, cell signaling, epigenetics and inflammasomes. Therefore, we conclude that H_2_S is involved in many important cellular and physiological processes. Compounds that donate H_2_S to biological systems can be developed as therapeutics for different diseases.

## 1. Introduction

Hydrogen sulfide (H_2_S) was first discovered as a toxic gas and an environmental pollutant three centuries ago [1,2,3]. Exposure to high concentrations of H_2_S for a long periods of time causes neurological, cardiovascular and pulmonary symptoms, which may eventually lead to death. However, it has also been noted that at low concentrations (<100 ppm), H_2_S triggers minimal clinical impairment [2]. Recently, a plethora of work has identified H_2_S as the third gasotransmitter, in addition to the two more heavily studied ones: nitric oxide (NO) and carbon monoxide (CO) [4,5]. Several enzymes, such as cystathionine-β-synthase (CBS), cystathionine-γ-lyase (CTH, also known as CSE) and 3-mercapto-sulfurtransferase (MPST, also known as 3-MST) were identified in mammals that directly or indirectly metabolize L-cysteine and produce endogenous H_2_S. H_2_S is mostly synthesized by these H_2_S synthases in the cytosol. However, hypoxia and other stressors can trigger translocation of these H_2_S synthases into mitochondria to generate H_2_S, where the Cys concentration is three-fold higher than that of the cytosol [6]. The discovery of these endogenous H_2_S synthases further confirms that H_2_S exists inside mammals and likely plays important physiological functions, such as cell differentiation, development, cardioprotection, vasodilation and immune responses [1].

In addition to endogenous H_2_S, exogenous sources from the consumption of natural H_2_S-producing compounds, such as those present in vegetables, or the administration of synthetic H_2_S donors can achieve similar physiological effects [7]. Currently, studies are ongoing toward the development of H_2_S donors as therapeutic drugs for various diseases [8,9,10,11]. Among them, sulfide salts (Na_2_S and NaHS) are considered fast H_2_S donors. Once dissolved in aqueous solution, sulfide salts release large amounts of H_2_S within a few seconds, which does not resemble the physiological condition [12,13]. Many slow-releasing H_2_S donors have been developed [8,9]. GYY4137 (Figure 1) has become the most widely used slow-releasing H_2_S donor in research, because it is commercially available and simple to handle [9]. H_2_S is slowly released by hydrolysis when GYY4137 is dissolved in a neutral aqueous solution [14]. GYY4137 was originally described as an accelerant to harden natural rubber. In 2008, GYY4137 was re-discovered as a slow-releasing H_2_S compound that can maintain low blood pressure by causing vasodilation and suppressing hypertension two weeks after in vivo injection in rats [12].

H_2_S has been the topic of numerous review articles due to its promising therapeutic potential for many different diseases. H_2_S donors can be divided into two major groups: (1) naturally occurring donors from food, such as diallyl disulfide (DADS) from garlic and oinons; (2) synthetic donors. Synthetic donors can be further divided into two subgroups: fast-releasing donors, such as Na_2_S and NaHS, and slow-releasing donors, such as GYY4137. Based on disease scenarios, different delivery approaches can be utilized to apply H_2_S donors-of-choice to patients to reach optimal results. The readers may refer to a few recent reviews on the different types of H_2_S donors and their potential clinical applications [8,9,10,11,15]. To write this review, we performed a literature search on recent publications related to H_2_S donors and the molecular and cellular responses after the treatment of H_2_S donors in in vitro and in vivo disease models. We will first discuss the direct targets of H_2_S and then examine the different cellular functions impacted by H_2_S donors. Our review will help understand the possible mechanisms of how H_2_S donors regulate different cellular pathways and functions to exert their therapeutical effects.

## 2. The Direct Target of H_2_S

H_2_S is a small molecule with reducing capacity. The small size makes it possible for H_2_S to penetrate through cell membrane. The reducing capacity of H_2_S enables it to interact with many different molecules [1]. H_2_S exists in three forms (Figure 2): the disprotonated (H_2_S), monoanion (HS^−^) and dianion (S^2−^) forms. All three forms are collectively referred to as hydrogen sulfide [16]. Because the pK_a1_ is ~6.8 and pK_a2_ is > 12 at 37 °C, hydrogen sulfide most likely exists as HS^−^ at physiological condition (pH ~5–7.8, 37 °C). H_2_S as gas can easily diffuse across membranes, but HS^−^ may only be transported by anion channels [17].

### 2.1. Targeting ROS/RNS and Forming Reactive Species Interactome (RSI)

The concept of reactive oxygen species (ROS) and reactive nitrogen species (RNS) has been widely explored. ROS include many derivatives of oxygen produced in the normal physiological process, such as hydrogen peroxide (H_2_O_2_) and superoxide (O_2_^−^). At a low steady-state level, ROS contributes to many normal cellular processes, including cell proliferation, differentiation and migration. However, too many ROS cause oxidative stress and result in inflammation, apoptosis and tumor growth [18]. Similarly, nitric oxide reacts with superoxide (O_2_^−^) and forms different derivatives called reactive nitrogen species (RNS), such as peroxynitrite (ONOO^−^). RNS production in macrophages and neutrophils is important for their anti-microbial function in eliminating pathogens, but overproduction of RNS can be harmful and result in tissue damage [19].

H_2_S, as a reducing agent, also reacts with ROS and RNS and participate in redox signaling. For example, H_2_S can directly interact with peroxynitrite (ONOO^–^) and produce ·HSO and ·NO [6,16]. It is suggested that H_2_S may be oxidized either in the mitochondria or via metal-catalyzed H_2_S oxidation [6,16].

Although H_2_S can directly interact with ROS and act as an ROS scavenger, this is probably a very minor role of H_2_S in most living cells [20,21]. The more important function of H_2_S in removing ROS is to regulate the expression of ROS scavengers, such as superoxide dismutase (SOD). H_2_S binds to SOD and enhances the rate of superoxide anion scavenging [22].

H_2_S can also suppress ROS indirectly by preserving antioxidants. There are many antioxidants that can counterbalance ROS, such as cellular reduced glutathione (GSH), GSH/GSSG ratio and thioredoxin (Trx-1). These cellular non-enzymatic antioxidants in H_2_S-treated cells can be maintained for long periods to counteract ROS, even though H_2_S no longer exists in the cultured media. Furthermore, H_2_S can reduce cystine to cysteine and facilitate GSH synthesis, indirectly suppressing ROS activity [21]. 

In fact, it is believed that H_2_S forms reactive sulfur species (RSS) that played a critical role in redox signaling when the earth had more sulfur than oxygen in the atmosphere about a billion years ago [6,23]. RSS, together with ROS and RNS, form reactive species interactome (RSI), mediating complex signaling pathways in cells. The RSI is robust and flexible, allowing efficient sensing and adaptation to environmental changes to enhance fitness and resilience [23].

### 2.2. Binding to HEME Proteins

H_2_S can directly interact with the amino acid histidine in heme-containing proteins, such as hemoglobin, and form sulfheme complexes in many invertebrates living in sulfide-rich environments [24]. In vertebrates, sulfide-ferric heme complexes may provide the storage, transport and detoxification of H_2_S [25,26]. H_2_S directly binds to the ferric heme structure in Cytochrome *c* oxidase (CcO), the last enzyme in the respiratory electron transport chain in mitochondria, and modifies its activity [27]. At low concentration (1:1 stoichiometry), H_2_S acts as a substrate to CcO without inhibiting its activity. At moderate to high concentration (1:2–3 stoichiometry), H_2_S inhibits CcO, resulting in the well-known toxicity of H_2_S [26]. Amino acids at the distal site in hemeproteins are important in stabilizing H_2_S-ferric binding. For example, in ferrous myeloperoxidase, a common distal arginine may stabilize the binding of sulfide [25]. In vertebrates, H_2_S also binds to oxygenated hemoglobin and myoglobin, which forms sulfhemoglobin and sulfmyoglobin. These sulfheme protein complexes are essential to H_2_S degradation in vivo [24]. Although H_2_S binding to hemeproteins is expected to be essential to many biological functions, little is known regarding the sulfide-heme binding mechanisms. The physiologic functions of sulfide-hemeprotein complexes still need to be further addressed [24,25].

### 2.3. Persulfidation

One type of post-translational modification of proteins is persulfidation (S-sulfhydration). This occurs by adding a sulfur molecule, usually from oxidized H_2_S, to an existing thiol (-SH) group of a cysteine residue [28,29]. Protein persulfidation can either stabilize and enhance its activity or inhibit its function [16]. Gao et al. developed a proteomics approach to quantitatively measure S-sulfhydrated cysteines. They found that elevated H_2_S promoted the persulfidation of metabolic enzymes and stimulated glycolytic flux in pancreatic β cells experiencing endoplasmic reticulum (ER) stress [30]. NaHS treatment persulfidated the Kir6.1 subunit of the KATP channel at Cys43 and enhanced the binding of Kir6.1 to the signaling molecule PIP2, leading to KATP channel activation [31]. H_2_S also persulfidated and inactivated phosphatase PTP1B, playing an important role during the response to ER stress [32]. Protein disulfide isomerase (PDI) is a key enzyme of protein folding in the ER. Jiang et al. showed that NaHS and GYY4137 treatments enhanced PDI activity, which was inhibited by mutating four cysteines in potential sulfhydrated sites of PDI [33].

Endothelial nitric oxide synthase (eNOS) is one of the three isoforms of NOS enzymes which synthesizes nitric oxide in endothelial cells. eNOS exists in either monomer or dimer formats. The dimeric eNOS is considered to be the active form [34]. NaHS treatment enhanced the persulfidation of eNOS at Cys443 and stabilized eNOS dimers, resulting in enhanced eNOS activity [35]. Nrf2 (Nuclear factor-erythroid factor 2-related factor 2) is an important transcriptional factor that regulates the expression of antioxidant proteins. Kelch-like ECH associating protein 1 (Keap1) binds to Nrf2 and acts as a negative regulator of Nrf2 [36]. NaHS treatment caused persulfidation in Keap1 at cysteine-151, resulting in Nrf2 dissociation from Keap1. Dissociated Nrf2 then translocated to the nucleus and enhanced the mRNA expression of Nrf2-targeted downstream genes [37]. P66Shc, an upstream activator of mitochondrial redox signaling, was persulfidated at Cysteine-59 when treated with NaHS. This attenuated the function of p66Shc and resulted in diminished mitochondrial ROS production [38]. 

### 2.4. NO/CO Feedback Loop 

These three gasotransmitters, NO, CO and H_2_S, also directly and indirectly interact with each other to regulate the signaling pathways. For example, H_2_S can persulfidate eNOS and facilitate NO production, as mentioned above in 2.3 [35]. Heme oxygenase-1 (HO-1) and HO-2, endogenous CO synthases, are Nrf2 downstream targeting genes. H_2_S can persulfidate Keap1 Cys151 and activate the Keap1/Nrf2 system, as mentioned in 2.3 [39]. On the other hand, both NO and CO can bind to the heme sites of H_2_S synthase, CBS, and modify its function in vivo, impacting H_2_S homeostasis [40]. As mentioned in 2.1, NO and H_2_S can directly bind to each other, resulting in GSNO, HSNO/ONS. Therefore, adding exogenous H_2_S will affect the redox states of the cells and modify NO and CO signaling. 

In conclusion, H_2_S can directly bind to small molecules and macromolecules to deliver its physiological function in vivo.

## 3. H_2_S Regulates Different Cellular Processes and Functions

H_2_S has the capacity to directly target many important signaling molecules and functional proteins. As a result, it plays a key role in many cellular processes and functions. Recent works have suggested that H_2_S donors can modify many cellular processes and exert therapeutic effects.

### 3.1. Cell Signaling Pathways

#### 3.1.1. PI3K/Akt Signaling Pathway

Many investigations have shown that H_2_S can activate the PI3K/Akt signaling pathway, which is highly conserved and tightly regulated among almost all cells and tissues. Phosphoinositide 3-kinase (PI3K) is activated by extracellular stimuli through membrane-bound receptors, such as receptor tyrosine kinase (RTK), which is a G-protein-coupled receptor (GPCR). Activated PI3K phosphorylates the 3′ hydroxyl of the inositol head group of phosphoinositides PIP2 and produces the lipid second messenger PIP3 [41]. Akt, also called protein kinase B (PKB), is an important serine/threonine kinase. Akt can be activated in a PI3K-dependent and independent manner. As the downstream mediator of PI3K, the lipid products (PI3, 4P2 and PIP3) of PI3K directly bind to the pleckstrin homology (PH) domain of Akt to activate this serine/threonine kinase. Akt activity can also be regulated by other proteins. For example, Akt can be activated by PIP3-dependent protein kinase 1 (PDK1) phosphorylating T308 of Akt. Phosphatases, such as PP2A and PHLPP, can dephosphorylate Akt at T308 and S473, to inactivate Akt. PTEN also suppresses Akt activity by dephosphorylating the lipid second messenger PIP3 [42]. Activated Akt can directly phosphorylate many downstream targets, including transcription factors, protein kinases, cell cycle regulators, etc. The Forkhead box O 1 (FOXO1) transcription factor is one of the well-studied targets of Akt. FOXO1 is considered a repressor of adipogenesis and insulin stimulated genes. Once phosphorylated by Akt, FOXO1 binds to 14-3-3 family of phospho-binding proteins and is transported from the nucleus to the cytosol. By removing FOXO1 from the nucleus, Akt activates adipogenesis gene expression [42] (Figure 3).

High glucose in diabetic mice and cell culture is associated with increased levels of PHLPP-1 and reduced p-Akt. GYY4137 pre-treatment prevented change in the expression of PHLPP-1 and p-Akt [43]. In a myocardial ischemia/reperfusion rat model, GYY4137 treatment protected the heart and increased p-Akt. This effect was abrogated by co-treating with PI3K inhibitor LY294002. These results suggested that GYY4137 exerted cardioprotective effects by activating the PI3K/Akt signaling pathway [44]. In an ipsilateral epididymis injury rat model, GYY4137 alleviated sperm damage and epididymis injury by the activation of the PI3K/Akt pathway [45].

In a liver ischemia/reperfusion (I/R) injury rat model, GYY4137 and/or NaHS treatment elevated the plasma levels of H_2_S, upregulated miR-21 and resulted in activation of the Akt pathway. Blocking miR-21 prevented the protection from NaHS by inactivating the Akt pathway, suggesting that miR-21 was important in mediating the protective effects of H_2_S in I/R-induced liver injury [46].

In an aortic cross-clamping rat model that induces acute lung injury, GYY4137 increased p-Akt and the downstream factors GSK3β and S6K and attenuated the increase of Angiopoietin-2 (Ang2, promotes cell death and disrupting vascularization) in lung tissue, caused by aortic cross-clamping [47].

Homocysteine (HHcy) treatment induces apoptosis (more details in Section 3.4) and matrix remodeling in mesangial cells. HHcy inactivates Akt and therefore activates the downstream FOXO1, resulting in the induction of apoptosis and the synthesis of excessive ECM protein. However, GYY4137 prevented the dephosphorylation of Akt and increased the nuclear translocation of FOXO1 in HHcy-treated cells, indicating that H_2_S regulates the Akt/FOXO1 pathway [48].

In a rat stomach culture model, GYY4137 increased phosphorylation of Akt and suppressed the secretion of the hunger hormone Ghrelin [49]. In apolipoprotein E KO mice, GYY4137 activated the PI3K/Akt pathway and reduced TLR4 expression, which was critical in preventing atherosclerotic plaque formation [50]. 

#### 3.1.2. NF-κB and MAPK Signaling Pathways

Besides the PI3K/Akt signaling pathway, NF-κB and MAPK pathways are also shown to be impacted by H_2_S treatment. Nuclear Factor-κB (NF-κB) is a family of inducible transcription factors that regulates a myriad of genes involved in inflammatory and immune responses. There are five main members in the NF-κB family: NF-κB1 (also named p50), NF-κB2 (also named p52), RelA (also named p65), RelB and c-Rel. In normal conditions, NF-κB are inactive and suppressed by a family of inhibitory proteins, such as IκB. Upon stimulation, a multi-subunit IκB kinase (IKK) complex is formed. IKK phosphorylates and degrades IkB proteins trigger the activation of NF-κB proteins [51,52] (Figure 4).

Cisplatin is a chemotherapy agent which can cause extensive nephrotoxicity. In cisplatin-treated renal tubular cells, H_2_S donors (NaHS and GYY4137) persulfidated STAT3 and IKKβ, resulting in the suppression of an NF-κB-mediated inflammatory cascade, which could potentially ameliorate renal damage caused by chemotherapy [53]. In murine macrophage RAW264.7 cells, LPS treatment increased the p-IκB and nuclear translocation of p65. The addition of GYY4137 suppressed the upregulation and therefore inhibited the LPS-induced activation of RAW264.7 cells [54].

The mitogen-activated protein kinase (MAPK) cascades are a chain of conserved proteins that communicate extracellular stimuli from the cell surface all the way to the DNA in the nucleus, resulting in cell proliferation, differentiation, motility and apoptosis [55]. There are three MAPKs: ERK, JNK and p38. Upon activation, these MAPKs are activated by their upstream kinases, MAP4K, MAP3K and MAPKK. Activated MAPKs phosphorylate their downstream substrates, mostly transcription factors, such as c-Myc, c-Jun, CREB, etc. [55,56] (Figure 5). Cisplatin induces the phosphorylation of ERK, JNK and p38 MAPKs, causing cell apoptosis. H_2_S donors NaHS and GYY4137 attenuated cisplatin-induced cell death by suppressing MAPK activation caused by cisplatin treatment. Cisplatin also suppressed H_2_S synthase CSE and CBS. The overexpression of CSE provided resistance to cisplatin-induced cell death. H_2_S donors also inhibited cisplatin-induced NADPH oxidase activation and p47phox phosphorylation, possibly by inducing the persulfidation of p47phox [57]. However, in the absence of mitogens like cisplatin, H_2_S donors themselves may act as mitogens and activate MAPK pathways. In a cell line that secretes glucagonlike peptide-1 (GLP-1), H_2_S donors NaHS and GYY4137 activated p38 MAPK and stimulated GLP-1 secretion [58]. In cultured mouse neurons, GYY4137 activated ERK but not p38 or JNK, resulting in the upregulation of acid-sensing ion channels (ASICs) [59].

### 3.2. Tight Junctions

Tight junctions are special structures in epithelia and endothelia that form barriers to separate different compartments. Tight junctions comprise transmembrane proteins (MARVEL-domain proteins, Claudins, E-cadherin, etc.) and a complex protein network in the cytosol called junctional plaque components (ZO1, ZO2, ZO3 and some other adaptor signaling proteins) that connect the transmembrane proteins to the cytoskeleton (microtubules and actin filaments). Tight junctions in epithelia and endothelia determine the body intactness. Mutations in tight junction proteins or tight junction disruption by microbial infections are linked to many diseases [60]. H_2_S slow-releasing donor GYY4137 treatment was shown to preserve the barrier function in tight junctions in mouse models of endotoxemia and sepsis [61,62]. Sodium deoxycholate (SDC), a bile acid salt, interrupted the tight junctions and increased phosphorylation of myosin light chain kinase (MLCK) and myosin light chain (MLC) in Caco-2 cells. The addition of GYY4137 to cultured Caco-2 cells suppressed the phosphorylation of MLCK and MLC induced by SDC. Mice that received SDC by oral gavage developed colitis. The intraperitoneal injection of GYY4137 in SDC-treated mice also preserved the integrity of the intestine and alleviated the colitis [63]. Zhao et al. demonstrated that GYY4137 treatment reduced intestinal permeability and upregulated tight junctions in a Dextran sulfate sodium (DSS)-induced mouse colitis model [64]. It is so far unknown what the direct targets of H_2_S are in enhancing tight junctions and preserving intestinal integrity.

### 3.3. Autophagy

When mammalian cells are under stress, such as infection, nutrient deprivation and hypoxia, they will start an adaptive process called autophagy to guarantee nutrients for vital cellular functions [65]. Autophagy is also essential in removing harmful cytosolic materials to prevent diseases such as neurodegeneration and cancer [65]. The process of autophagy typically starts with the recruitment of autophagy-related proteins (ATGs) to a specific subcellular location and the formation of a phagophore, a cup-shaped membrane structure. The phagophore keeps elongating and encircles cytosolic material to form a round double-membrane vesicle called autophagosome. Autophagosomes eventually fuse with lysosomes to form autolysosomes where the engulfed cytosolic material and autophagic body are degraded and recycled. In addition to ATGs, there are many other proteins participating in the process of autophagy, such as the serine/threonine protein kinase ULK1, ubiquitin, LC3s, etc. These autophagic proteins are highly regulated. Autophagy has been proven to be essential to cellular homeostasis by extensive research in recent years. Disorders in autophagy are linked to diseases such asneurodegeneration, immunological diseases and cancer, as well as aging [65].

In human umbilical vein endothelial cells (HUVECs), GYY4137 treatment enhanced autophagic flux by increasing LC3BII expression, exerting cytoprotective function. This process was shown to be dependent on Sirt1/FoxO1 pathway [66]. In an experimental periodontitis model, GYY4137 prevented excessive inflammation by inducing autophagy [67]. In a streptozotocin-induced diabetic model, NaHS treatment activated the AMPK/mTOR pathway and increased the levels of ATGs. Increased autophagic ultrastructures and LC3-I/LC3-II conversion were also observed in the NaHS-treated group, further confirming that H_2_S activated autophagy [68]. 

### 3.4. Apoptosis

Apoptosis is the process of programmed cell death, which can be induced either through intrinsic or extrinsic pathways. The release of Cytochrome *c* from mitochondria often intrinsically triggers the process of apoptosis, including the formation of apoptosomes and caspase 3 activation. BCL-2 family proteins are a group of conserved proteins that share Bcl-2 homology domains and are well known as the regulators of apoptosis. Some BCL-2 family proteins, such as BIM, BID, PUMA, BAX and BAK1, are pro-apoptotic, while others, such as BCL-2, BCL-Xl, BCL-W, BCL-2-A1 and MCL1, are anti-apoptotic. The expression and function of BCL-2 family proteins can be regulated by Cyclin-dependent kinases (CDKs) and p53. Apoptosis can also be induced by cell membrane proteins called death receptors, including Fas, TNFR1, TNFR2 and TRAIL receptors. Upon binding to their ligands, death receptors are activated, triggering the recruitment of adaptor proteins FADD and the caspase cascade. The activation of caspases leads to the cleavage of BID, the activation of pro-apoptotic BCL-2 family proteins and the release of Cytochrome *c* from mitochondria [69]. Many groups have demonstrated that H_2_S protects cells against apoptosis and/or promotes autophagy, as discussed in Section 3.3. Li et al. showed that GYY4137 treatment in cardiomyocytes attenuated high glucose-induced apoptosis by decreasing caspase-3 activity and pro-apoptotic BAX and increasing anti-apoptotic BCL-2. The effect was probably mediated via the inhibiting STAT3/HIF-1α pathway [70]. Cardiomyocyte HL-1 cells preconditioned with GYY4137 were protected from nutrient deprivation-induced apoptosis. Furthermore, GYY4137 pretreatment reduced the level of cleaved caspase-3 and preserved ATP in isolated rat heart [71]. In a pathological infertility rat model, as mentioned in Section 3.1, GYY4137 treatment increased the level of antioxidant superoxide dismutase (SOD), the phosphorylation of PI3K p85 and Akt. The treatment also suppressed the pro-apoptotic marker active caspase-3 and Bax, resulting in the alleviation of sperm damage [45]. In a neuropathic pain mouse model, pretreatment with GYY4137 and diallyl disulfide (DADS) improved the antiallodynic effects of opiate agonists morphine and UFP-512. GYY4137 and DADS treatment also suppressed the increased pro-apoptotic Bax expression in the medial septum of mice with neuropathic pain [72].

Type 2 diabetes is an independent risk factor for a failed lung transplant. Jiang et al. showed that GYY4137 treatment in a lung I/R/type 2 diabetic rat model suppressed lung cell apoptosis by using TUNEL staining. Their data suggested that GYY4137 treatment rescued SIRT1 expression, lung function, oxidative damage and inflammation. GYY4137′s effectiveness was likely from the activation of SIRT1 signaling pathways, as the SIRT1 inhibitor, EX527, was shown to abolish all these effects [73].

Dexamethasone (Dex), a glucocorticoid that treats inflammation, was found to induce osteoporosis (loss of bone mass). At the same time, Dex suppressed serum H_2_S and H_2_S-generating enzymes in the bone marrow in vivo. In Dex-treated rats, GYY4137 alleviated osteoporosis by rescuing cell proliferation and the expression of signature proteins (i.e., Runx2, alkaline phosphatase) in osteoblasts, cells specialized in bone matrix synthesis and mineralization. GYY4137 also suppressed Dex-induced apoptosis by increasing the ratio of Bcl-2/Bax in osteoblasts [74].

The high level of plasma homocysteine is common in patients with chronic kidney diseases and the integrity of mesangial cells is essential to the function of kidney. Homocysteine induces apoptosis in cultured mouse mesangial cells. The addition of GYY4137 to the homocysteine-treated mesangial cells prevented apoptosis with reduced cleaved-Caspase 3. GYY4137 also blocked Akt/FOXO1 activation, ROS production, matrix protein induction and mitochondrial dysfunction [48]. In a diabetic cardiomyopathy rat model, GYY4137 also suppressed high-glucose-induced oxidative stress and apoptosis. However, in this model, GYY4137 induced FOXO1 phosphorylation [75], which is consistent with the previous discussion in Section 3.1.1, indicating that H_2_S donors promote the PI3K/Akt/FOXO1 pathway.

GYY4137 treatment prevented mitochondria-mediated apoptosis in both cultured cells and mice deficient in H_2_S-generating enzymes. Further experiments suggested that mitochondrial protein ATP synthase ATP5A1 could be persulfidated by H_2_S. When Cys244 in ATP5A1 was mutated into serine (C244S), GYY4137 no longer prevented mitochondria-mediated apoptosis, suggesting that persulfidation by H_2_S is a possible mechanism in regulating mitochondria function and cytoprotection [76]. Osteoarthritis is characterized by cartilage erosion. Chondrocytes, the only cell type present and essential to cartilage development, goes through apoptosis when under oxidative stress. Treatment with GYY4137 ameliorated chemical-induced ROS production and apoptosis in chondrocytes. GYY4137 also suppressed stress-induced mTOR and P70S6K activation in chondrocytes. GYY4137-treated chondrocytes showed an increased LC3II/LC3I ratio, implying GYY4137 also promoted autophagy in these cells [77].

### 3.5. Vesicle Trafficking: Exocytosis/Endocytosis/Pinocytosis

Vesicle trafficking is essential to the function of all live mammalian cells. Molecules are transported in between specific membrane-enclosed compartments inside the cell as well as between a cell and its environment to maintain the functional organization of cells. Vesicle trafficking can be divided into many different processes based on the direction of trafficking (endocytosis, exocytosis, transcytosis) as well as cargo specificity (pinocytosis, phagocytosis, clathrin-dependent and -independent endocytosis, COP-mediated vesicle transport, etc.) [78]. Here we will focus on three types of vesicle trafficking: exocytosis, endocytosis and pinocytosis.

Exocytosis is the process through which intracellular vesicles fuse with the plasma membrane and release their cargo to the extracellular environment. In excitable cells, such as neurons and endocrine cells, this process is highly regulated and the cells can release large amounts of neurotransmitters or hormones within milliseconds after being triggered [78]. In pancreatic β cells, glucose stimulation can trigger the influx of ions, a second messenger, and results in a signaling cascade leading to exocytosis and the release of insulin [79]. Chromaffin cells are neuroendocrine cells found mostly in the adrenal glands in mammals. Exogenous H_2_S can regulate intracellular calcium in chromaffin cells, and therefore, facilitate exocytosis and the release of the neurotransmitter catecholamines [80].

Endocytosis is the process that a wide range of cargo molecules can be transported from outside to the inside of the cells. Clathrin is a protein that coats vesicles and forms complexes with other adaptor/signal proteins to transport vesicles to specific compartments [81]. Endocytosis can be divided into well-studied clathrin-dependent endocytosis and less-studied clathrin-independent endocytosis. Clathrin-independent endocytosis utilizes cholesterol-rich membrane domains called lipid rafts and contains scaffold protein caveolins [82]. Pinocytosis, “cell drinking”, is a type of endocytosis that uptakes liquid content, such as lipid droplets. The size of vesicles in pinocytosis is very small and clear compared to those in other endocytosis [83]. NaHS treatment inhibited Na^+^/K^+^-ATPase activity by enhancing Na^+^/K^+^-ATPase endocytosis in rat renal tubular epithelial cells [84]. Further studies found that H_2_S directly bound and activated EGFR by persulfidation at Cys797 (human) or Cys798 (rat), triggering its downstream Gab1/PI3K/Akt pathway and enhanced endocytosis. An intravenous injection of NaHS increased water and sodium excretion in the rat kidney, suggesting H_2_S can be developed as therapeutics for diseases related to renal sodium homeostasis dysfunction [84]. In a different study, inhibiting H_2_S synthesis enhanced and NaHS treatment inhibited CXCR2 endocytosis in mouse neutrophils in a K(ATP)(+) channel-dependent mechanism [85].

### 3.6. Epigenetics

Epigenetics is a process that alters gene activity but does not change the DNA sequence, including but not limited to DNA methylation, chromatin modification and noncoding RNAs. Epigenetics can be regulated by genetic, developmental and environmental factors [86]. SIRT1 is a deacetylase and can deacetylate histones and modify chromatin structures [87]. Previous work has shown that H_2_S can increase SIRT1 expression. Furthermore, H_2_S can persulfidate SIRT1 and stabilize it. Therefore, H_2_S can enhance SIRT1 activity and promote deacetylation to regulate the expression of inflammatory proteins [88]. In HUVECs, H_2_S treatment enhanced the deacetylase activity of SIRT1, which induced the deacetylation of FOXO1 and the nucleus translocation of FOXO1 [66].

MicroRNAs are small single-stranded noncoding RNAs (~22 nucleotides) that function as post-transcriptional regulators of gene expression. Hundreds of different miRNAs have been discovered in humans and many of them are conserved in other mammals [89]. As mentioned in Section 3.1, H_2_S donor treatment elevated the plasma levels of H_2_S, enhanced microRNA-21 (miR-21) expression and activated the Akt pathway in a rat liver I/R model. Anti-miR-21 annulled the protective effects of H_2_S donor and inactivated the Akt pathway in the same model [46]. miR-129 was downregulated in a chronic kidney disease mouse model. The treatment of GYY4137 ameliorated the disease and rescued miR-129 expression in this model [90]. In a diabetic mouse model, there were decreased H_2_S in plasma, reduction in miR-194, collagen deposition and fibrosis in the diabetic kidney. GYY4137 treatment rescued plasma miR-194 expression and mitigated renal disease symptoms [91]. In cardiomyocytes, Na_2_S-treamtent upregulated miR-133a expression, which inhibited cardiomyocytes hypertrophy [92].

### 3.7. NLRP3 Inflammasome

Upon sensing pathogen-associated molecular patterns (PAMPs) or damage-associated molecular patterns (DAMPs) in the cytoplasm, proteins, such as the nucleotide-binding oligomerization domain (NOD) and leucine rich repeat (LRR)-containing receptors (NLRs), will form a multimeric protein complex with adaptor molecule ASC and the cysteine protease procaspase-1. Besides NOD and LRR domains, NLRs also contain pyrin domains which can bind to the pyrin domains of adaptor protein ASC. ASC has two domains: an N-terminal pyrin domain and a carboxy-terminal caspase recruitment domain (CARD), connecting NLRs to procaspase-1. Multiple NLRs, ASCs and procaspase-1s bind together and form a speck-like structure to induce the self-cleavage of procaspase-1, producing the active form caspase-1. Caspase-1 can cleave many precursors of proinflammatory cytokines, such as interleukin-1β, IL-18 and Gasdermin-D, triggering inflammation and pyroptosis (a form of cell death triggered by inflammatory signals) [93,94]. NLRP3 inflammasomes are the most studied PAPMs/ DAMPs sensing NLRs. Mutations in NLRP3 have been associated with autoinflammatory disorders [93,94].

Monosodium urate (MSU) crystals, the factor contributing to gout arthritis, activate the NLRP3 inflammasomes. Using mouse models and in vitro cell culture, researchers observed increased inflammation after treatment with MSU crystals, which resulted in a deficiency of CSE. The addition of GYY4137-ameliorated caspase-1 activity, ASC oligomerization, IL-1β secretion and inflammation, suggests that H_2_S inhibits NLRP3 inflammasome activity [95].

GYY4137 treatment suppressed NLRP3 expression both in vivo and in vitro using a diabetes-accelerated atherosclerosis model. Furthermore, GYY4137 treatment reduced plaque formation in vivo and ICAM1/VCAM1 levels in vitro [96]. In a mouse sepsis model, GYY4137 treatment reduced NLRP3 and caspase-1, and alleviated lung injury [97]. In a sepsis mouse model, GYY4137 treatment attenuated sepsis-induced cardiac dysfunction in WT but not Nlrp3 KO mice, suggesting that GYY4137 inhibited the inflammasome pathway. In fact, GYY4137 suppressed NLRP3 inflammasome activities in macrophages and reduced the infiltration of macrophages in septic heart tissue [98].

### 3.8. Ion Channels

There are protein complexes on the cell surface and in the intracellular membrane that form pores to allow ions (Na, K, Ca, Cl, etc.) to be transported. These complexes are called ion channels. Some ion channels are gated by voltages while others are voltage-insensitive and gated by other mediators [56]. H_2_S has been shown to regulate and modify ATP-sensitive potassium (KATP) channels and calcium channels. The KATP channel has eight subunits: four pore-forming Kir6.X subunits (either Kir6.1 or Kir6.2) and four regulatory sulfonylurea receptor (SUR) subunits. Kir6.X contains a highly conserved sequence that allows permeating K^+^ ion. It also has binding sites for ATP and phosphatidylinositol 4,5-bisphosphate (PIP2), which inhibits and activates KATP channel, respectively. SUR subunits contain nucleotide-binding sites, which also regulate the opening/closure of KATP channels [99]. KATP channels are common to many cell types but mostly studied in cardiomyocytes and pancreatic β-cells regulating metabolism [99]. Calcium channels are important to the function of the brain, heart and muscle. There are many different isoforms of calcium channels which perform multiple physiological functions [100]. H_2_S donors NaHS and GYY4137 persulfidated Kv2.1, a subunit in voltage-gated K (+) channels and inhibited its function in rat neurons [101]. In rat atria, treatment with NaHS or Na_2_S augmented stretch-induced atrial natriuretic peptide (ANP) secretion and this effect was blocked by pretreatment with a KATP channel inhibitor, suggesting that H_2_S stimulates ANP secretion via the KATP channel [102]. In a mouse paclitaxel-induced neuropathic pain model, GYY4137 treatment demonstrated anti-hyperalgesic activity and co-treatment with a KATP channel inhibitor blocked this anti-hyperalgesic activity [103]. In cultured carotid body, a cluster of chemoreceptor cells from bilateral sensory organs in the peripheral nervous system, NaHS treatment triggered transient calcium influx, which depended on the calcium in the culture media, suggesting that H_2_S regulates calcium channels in carotid bodies [104].

## 4. Conclusions

Our current knowledge suggests that H_2_S can directly persulfidate proteins and metabolites, and likely influence the NO/CO feedback loop. H_2_S modifies the metal-binding complex in hemeproteins to regulate their functions. It can also directly modulate ROS/RNS and affect redox signaling. These direct effects may impact cellular functions, such as regulating tight junctions, autophagy/apoptosis, signaling pathways, epigenetics and ion channels, resulting in physiological changes, such as vasodilation, cardioprotection, cell differentiation, etc. (Figure 6). However, more work is needed to precisely identify the working mechanisms of H_2_S as a gasotransmitter in mammals. For instance, methods for the precise measurement of H_2_S concentration in different tissues are still lacking [1,105]. Although a useful tool in studying H_2_S function, GYY4137 has a few drawbacks. For example, GYY4137 releases byproducts other than H_2_S after hydrolysis, which may contribute to the observed physiological function in vivo and in vitro [9]. So far, no H_2_S donor has been approved in clinical trials. This is partly due to the lack of a reliable method measuring free H_2_S in human samples, resulting in difficulties in studying the pharmacokinetics of H_2_S donors [106]. Nonetheless, H_2_S functioning as a neuromodulator was only first reported in the 1990s [5] and as a gasotransmitter in 2001 [107]. With promising data from animal disease models and more research emerges, H_2_S donors have great potential as future therapeutic agents for many cardiovascular, neurological and inflammatory diseases.

## Figures and Tables

**Figure 1 antioxidants-11-01788-f001:**
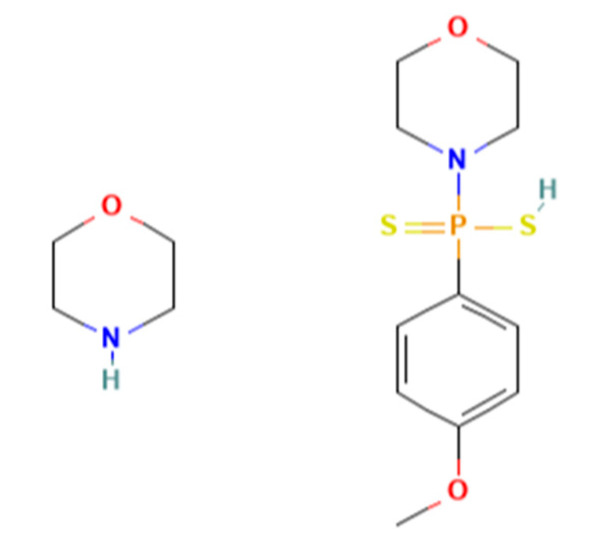
Chemical structure of GYY4137.

**Figure 2 antioxidants-11-01788-f002:**
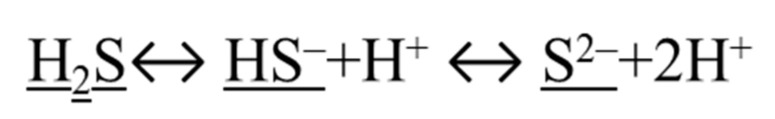
Three forms of H_2_S are underlined.

**Figure 3 antioxidants-11-01788-f003:**
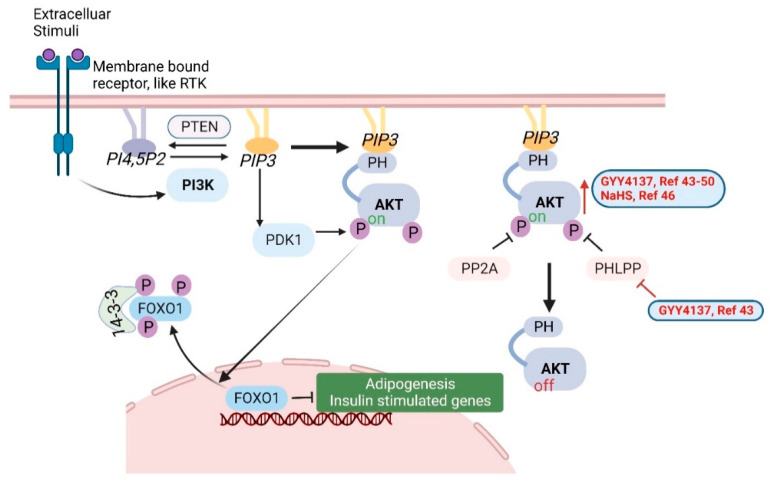
The PI3K/Akt signaling pathway. H_2_S donors (GYY4137 and NaHS) increase Akt activation, as shown in red font, arrow and lines [43,44,45,46,47,48,49,50]. Created with BioRender.com.

**Figure 4 antioxidants-11-01788-f004:**
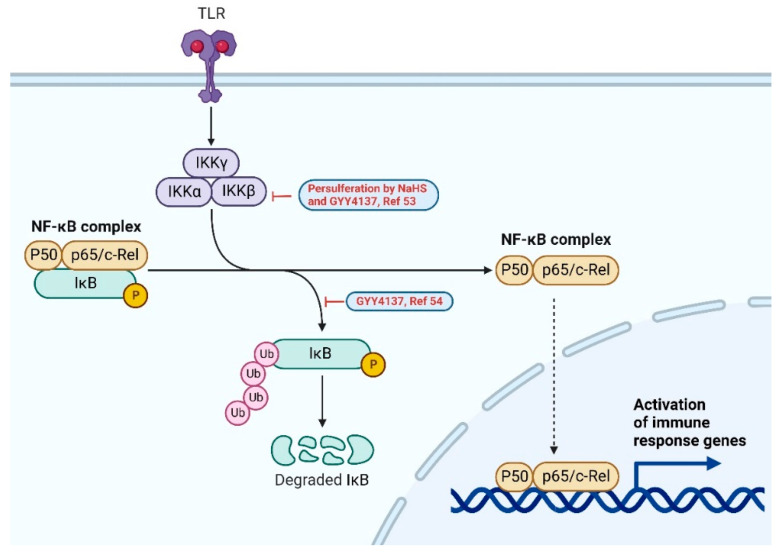
NF-κB signaling pathway. H_2_S donors (GYY4137 and NaHS) suppress the upregulation of phospho-IκB, as shown in red, and LPS-induced inflammation [53,54]. Created with BioRender.com.

**Figure 5 antioxidants-11-01788-f005:**
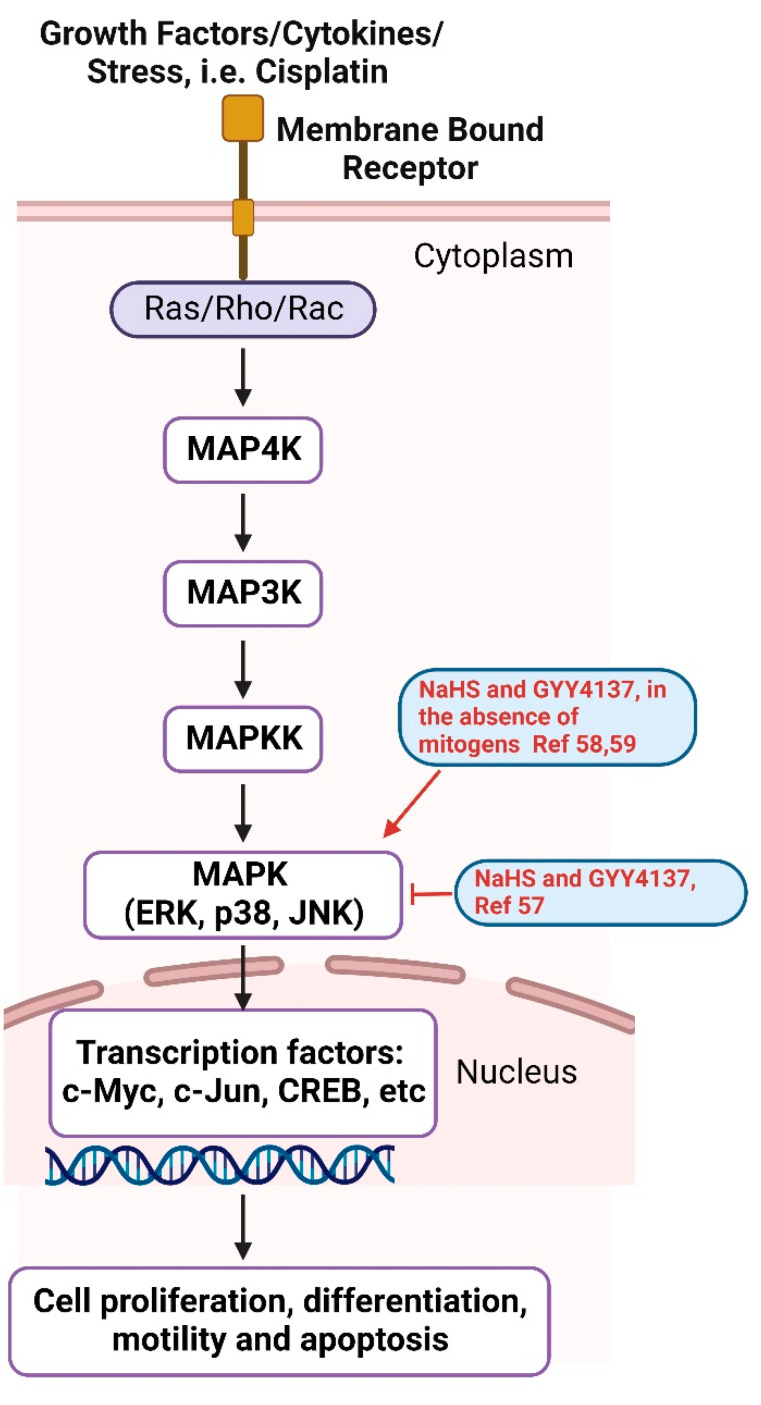
MAPK signaling pathway. H_2_S donors (NaHS and GYY4137) regulate MAPK activation, as shown in red [57,58,59]. Created with BioRender.com.

**Figure 6 antioxidants-11-01788-f006:**
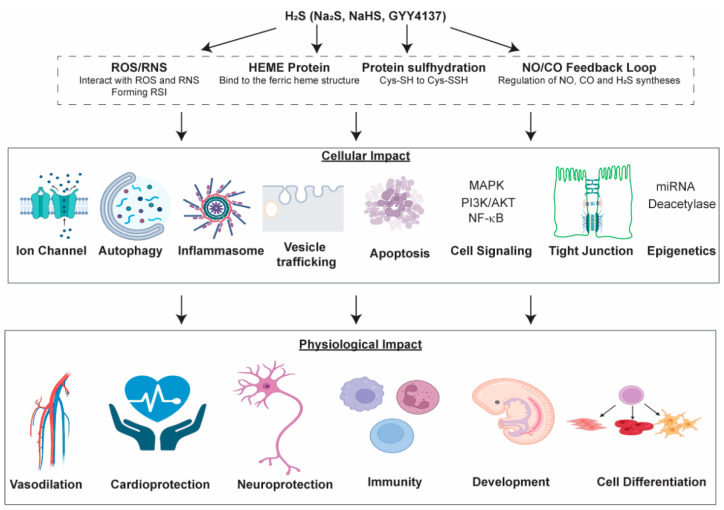
Illustration of the direct targets of H_2_S and the downstream cellular/physiological impacts.

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
