# Peer review of "Recent Development of the Molecular and Cellular Mechanisms of Hydrogen Sulfide Gasotransmitter"

_antioxidants, 2022, doi:10.3390/antiox11091788_

Round 1
Reviewer 1 Report
The manuscript reviewed the possible direct targets of H2S and listed research examples of using H2S donors to treat various diseases. This work is clearly structured with sufficient content. I would like to recommend its publication after addressing the following issues.
1. The full name of abbreviations should be provided upon the first appearance, e.g. H2S (Line 11).
2. The authors mentioned that direct modifications by H2S have significant impact on many cellular processes, such as tight junctions, autophagy, apoptosis, vesicle trafficking, cell signaling, epigenetics and inflammasomes (Line 17). We are wondering if it is reasonable to classify tight junctions and inflammasomes as cellular processes; please check it.
3. H2S donors (NaHS and GYY4137) were reported to attenuate or activate MAPK; please explain the reasons.
4. More up-to-date studies published in 2021 and 2022 were recommended to added to show the latest development in the field.
5. For which reasons no H2S donor has been approved in clinical trials? What are the limits to this? Besides NaHS and GYY4137, is there other H2S donor? When selecting a H2S donor for a certain disease, what should be taken into consideration?
6. Numerous reviews regarding the role of H2S gasotransmitter in diverse diseases have been reported. Please state the distinctions of this work compared with other works.
Author Response
Comments from Reviewer 1
The manuscript reviewed the possible direct targets of H2S and listed research examples of using H2S donors to treat various diseases. This work is clearly structured with sufficient content. I would like to recommend its publication after addressing the following issues.
- Q: The full name of abbreviations should be provided upon the first appearance, e.g. H2S (Line 11).
A: We thank the reviewer for pointing out this issue. We have edited the text accordingly in line 11 and 26.
- Q: The authors mentioned that direct modifications by H2S have significant impact on many cellular processes, such as tight junctions, autophagy, apoptosis, vesicle trafficking, cell signaling, epigenetics and inflammasomes (Line 17). We are wondering if it is reasonable to classify tight junctions and inflammasomes as cellular processes; please check it.
A:The authors agree that tight junctions are part of cell structure and inflammasomes belong to the function of cells. Therefore, we have edited the text by adding “cell structure and functions” to line 18, 177, 179.
- Q: H2S donors (NaHS and GYY4137) were reported to attenuate or activate MAPK; please explain the reasons.
A: We examined the related publications. The discrepancy is probably caused by different experimental systems. In Ref. 54, cisplatin worked as a mitogen and induced activation of MAPK, and H2S donors attenuated MAPK activation. But in Ref. 55 and 56, in the absence of mitogens, H2S donors activated p38 or ERK. The text has been edited accordingly in line 265-269.
- Q: More up-to-date studies published in 2021 and 2022 were recommended to added to show the latest developmentin the field.
A: We have added 9 citations and almost all are papers from 2021 and 2022.
- Q: For which reasons no H2S donor has been approved in clinical trials? What are the limits to this? Besides NaHS and GYY4137, is there other H2S donor? When selecting a H2S donor for a certain disease, what should be taken into consideration?
- Q: Numerous reviews regarding the role of H2S gasotransmitter in diverse diseases have been reported. Please state the distinctions of this work compared with other works?
A: We would like to answer Q5-6 together. Based on the information provided on clinicaltrial.gov, the limitations and caveats of clinical studies of H2S donors include a technical problem detecting free H2S levels in human samples. There are many naturally occurring and synthetic H2S donors described in many other high-quality reviews. Based on the disease scenarios, different delivery approaches may be used to achieve optimized clinical outcomes. However, these are not the focus of this current review. Instead, we would like to convene the information on the potential downstream mediators of H2S donors in mammalian cells and tissues to peek into possible acting mechanisms of H2S donors. The discussion has been added to this manuscript in line 58-74 and 504-506.
Reviewer 2 Report
Liu and coworkers review how H2S interacts with ROS/RNS, hemoproteins, metal-containing complexes etc. The authors show these interactions have significant impact on many processes such as tight junctions, autophagy, apoptosis etc. I am supportive of publication of this manuscript with a few clarifications listed below:
1. In Introduction, reference 7 and 8 are relatively old as the field of H2S donors has been developing very fast, new review articles should be cited.
2. Figure 1 legend writes H2S generated by hydrolysis but the figure doesn’t show the hydrolysis at all, I suggest the authors either not mentioning hydrolysis or drawing out the hydrolysis.
3. In multiple figures, for example, figure 3, please specify which H2S donor was studied?
Author Response
Comments from Reviewer 2
Liu and coworkers review how H2S interacts with ROS/RNS, hemoproteins, metal-containing complexes etc. The authors show these interactions have significant impact on many processes such as tight junctions, autophagy, apoptosis etc. I am supportive of publication of this manuscript with a few clarifications listed below:
- Q: In Introduction, reference 7 and 8 are relatively old as the field of H2S donors has been developing very fast, new review articles should be cited.
A: We thank the reviewer for bring up this great point. We have added two recent reviews in addition to several citations from 2021 and 2022 to this manuscript.
- Q: Figure 1 legend writes H2S generated by hydrolysis but the figure doesn’t show the hydrolysis at all, I suggest the authors either not mentioning hydrolysis or drawing out the hydrolysis.
A: We have deleted hydrolysis in Figure 1 legend.
- Q: In multiple figures, for example, figure 3, please specify which H2S donor was studied?
A: H2S donor names have been added to all figures.
Reviewer 3 Report
The manuscript "Recent development of the molecular and cellular mechanisms of hydrogen sulfide gasotrasmitter" by Liu et al. makes a comprehensive revision of the literature regarding the role of H2S molecule as a signalling molecule in mammals. Moreover, the authors provide a very interesting and complete overview of possible (in)direct targets of this gasotransmitter in vivo and present and discuss several downstream physiological impacts of these interactions.
The manuscript is written in a very clean and easy-to-read English, and its structure is very well schemed and easy to follow. This Reviewer congratulates the authors on the present manuscript and has only very (very) little observations.
1. on page 1, line 37-38, the authors refer to "H2S-synthases in the cytosol"; later on, on page 4, line 152, the authors mention the "H2S-synthase, CBS"; following the text on lines 31-40, it is not clear that the authors are referring to CBS, CSE and MST as H2S-synthases, hence, this Reviewer suggests the addition of "these" to the phrase making it: "H2S is mostly synthesized by THESE H2S synthases in the cytosol";
2. Figure 3 would gain if the "FOXO1" and "Adipogenesis Insulin stimulated genes" panels are not in red; when one looks to the Figure, these panels dominate rather than the panels which one should be drawn attention to (namely the ones with red fonts); this is in contrast to Figure 4 and 5 which, with no red panels, one is immediately drawn to where the authors which us to look;
3. for sake of completeness, these very interesting reviews could be mentioned en passant on the manuscript: PMID 35435014, 35618601 and 29112440;
4. the most commonly used nomenclature for CcO is "cytochrome c oxidase" (with the "c" italicized);
5. the most commonly used nomenclature for "pancreatic beta cells" is with the "beta" as the greek letter.
This Reviewer congratulates the authors on a great manuscript and wishes them all the best for their future endeavours.
Author Response
Comments from Reviewer 3
The manuscript "Recent development of the molecular and cellular mechanisms of hydrogen sulfide gasotrasmitter" by Liu et al. makes a comprehensive revision of the literature regarding the role of H2S molecule as a signalling molecule in mammals. Moreover, the authors provide a very interesting and complete overview of possible (in)direct targets of this gasotransmitter in vivo and present and discuss several downstream physiological impacts of these interactions.
The manuscript is written in a very clean and easy-to-read English, and its structure is very well schemed and easy to follow. This Reviewer congratulates the authors on the present manuscript and has only very (very) little observations.
1.Q: on page 1, line 37-38, the authors refer to "H2S-synthases in the cytosol"; later on, on page 4, line 152, the authors mention the "H2S-synthase, CBS"; following the text on lines 31-40, it is not clear that the authors are referring to CBS, CSE and MST as H2S-synthases, hence, this Reviewer suggests the addition of "these" to the phrase making it: "H2S is mostly synthesized by THESE H2S synthases in the cytosol";
A: We thank the reviewer for the comments. “these” is added to the text.
- Q: Figure 3 would gain if the "FOXO1" and "Adipogenesis Insulin stimulated genes" panels are not in red; when one looks to the Figure, these panels dominate rather than the panels which one should be drawn attention to (namely the ones with red fonts); this is in contrast to Figure 4 and 5 which, with no red panels, one is immediately drawn to where the authors which us to look;
A: Figure 3 has been modified accordingly.
- Q: for sake of completeness, these very interesting reviews could be mentioned en passanton the manuscript: PMID 35435014, 35618601 and 29112440;
A: These three reviews have been cited in the manuscript.
- Q: the most commonly used nomenclature for CcO is "cytochrome coxidase" (with the "c" italicized);
A: The “c” is now italicized.
- Q: the most commonly used nomenclature for "pancreatic beta cells" is with the "beta" as the greek letter.
A: The “beta” has been changed to “β”.
Reviewer 4 Report
This is a very thorough review on the effects of sulfide on cellular signaling pathways.
One major defficiency relates to persulfidation of thiols by H2S. It has been well documented that H2S does not directly persulfidate protein thiols. H2S must first be oxidized to persulfide before it reacts with free thiols. The abstract and the persulfidation section must be modified to reflect this fact.
Author Response
This Reviewer congratulates the authors on a great manuscript and wishes them all the best for their future endeavours.
This is a very thorough review on the effects of sulfide on cellular signaling pathways.
One major defficiency relates to persulfidation of thiols by H2S. It has been well documented that H2S does not directly persulfidate protein thiols. H2S must first be oxidized to persulfide before it reacts with free thiols. The abstract and the persulfidation section must be modified to reflect this fact.
A: We thank the reviewer’s compliments. The need of oxidant in H2S’s persulfidation function has been added to the abstract (line 16-17) and main text (line 139).